# CHUNKLLM: A LIGHTWEIGHT PLUGGABLE FRAMEWORK FOR ACCELERATING LLMS INFERENCE

## ABSTRACT

Transformer-based large models excel in natural language processing and computer vision, but face severe computational inefficiencies due to the self-attention's quadratic complexity with input tokens. Recently, researchers have proposed a series of methods based on block selection and compression to alleviate this problem, but they either have issues with semantic incompleteness or poor training-inference efficiency. To comprehensively address these challenges, we propose ChunkLLM, a lightweight and pluggable training framework. Specifically, we introduce two components: QK Adapter (Q-Adapter and K-Adapter) and Chunk Adapter. The former is attached to each Transformer layer, serving dual purposes of feature compression and chunk attention acquisition. The latter operates at the bottommost layer of the model, functioning to detect chunk boundaries by leveraging contextual semantic information. During the training phase, the parameters of the backbone remain frozen, with only the QK Adapter and Chunk Adapter undergoing training. Notably, we design an attention distillation method for training the QK Adapter, which enhances the recall rate of key chunks. During the inference phase, chunk selection is triggered exclusively when the current token is detected as a chunk boundary, thereby accelerating model inference. Experimental evaluations are conducted on a diverse set of long-text and short-text benchmark datasets spanning multiple tasks. ChunkLLM not only attains comparable performance on short-text benchmarks but also maintains 98.64% of the performance on long-context benchmarks while preserving a 48.58% key-value cache retention rate. Particularly, ChunkLLM attains a maximum speedup of 4.48× in comparison to the vanilla Transformer in the processing of 120K long texts.

## 1 INTRODUCTION

Transformer-based large models (Vaswani et al., 2017) have demonstrated exceptional performance across a diverse range of tasks, including natural language processing (Srivastava et al., 2025; Zhang et al., 2024) and computer vision (Jiang et al., 2025). However, they have also faced significant challenges in terms of computational efficiency, particularly when scaling to larger structures and large context inputs. A core issue of efficiency limitations lies in the self-attention module, whose computational complexity is a quadratic relationship with the number of input tokens. Such deficiencies in computational efficiency exert a profound impact on both the training complexity and inference latency of large models.

Efficiency optimization of Transformer has emerged as a pivotal research domain, with efforts predominantly converging into three methodological paradigms. **Linear attention**, such as Mamba (Dao & Gu, 2024), RWKV (Peng et al., 2023b; 2024), and RetNet (Sun et al., 2023), seek to approximate and substitute the traditional softmax-based self-attention mechanism. However, the fundamental architectural disparities between linear attention and conventional attention mechanisms introduce non-trivial challenges: adapting pre-existing Transformer models to integrate linear attention often incurs prohibitive conversion costs(Mercat et al., 2024; Wang et al., 2024; Bick et al., 2024), while alternative strategies necessitate end-to-end training of entirely new model from scratch(Li et al., 2025). Another optimization paradigm is **Sparse attention**, which leverages predefined structural constraints, such as sink-based attention mechanisms (Xiao et al., 2024) or sliding window attention mechanisms (Beltagy et al., 2020b), to exploit this sparsity. While these methods may yield certain effects, they often rely heavily on specific tasks, which can limit the overall

generalization ability of the model. Dynamic sparse attention mechanisms (Tang et al., 2024; Jiang et al., 2024; Liu et al., 2024) filter out subsets of tokens during the inference phase. Although such methods can reduce the computational load of long sequences, they fail to significantly lower the high training costs of long-context models, making it difficult for large language models to efficiently scale to context-processing tasks with million-level token sizes. **Chunk Selective attention**, a special type of sparse attention, can be primarily categorized into two paradigms: fixed chunk (Lu et al., 2025; Yuan et al., 2025; Wang et al., 2025) and separators-based dynamic chunk (Chen et al., 2024). Both approaches partition the input into discrete chunks: the former conducts partitioning with a fixed length, which gives rise to semantic incompleteness; the latter utilizes separators for partitioning, yet ambiguities often emerge. For example, periods frequently occur in numerical values or abbreviations. Furthermore, during the inference phase, these methods necessitate chunk selection for each generated token, incurring additional computational overhead. It is thus evident that existing efficient approaches still exhibit inherent limitations.

To address the aforementioned challenges, We propose ChunkLLM, which can be directly constructed by integrating two lightweight and trainable modules into existing LLMs: **QK Adapter** and **Chunk Adapter**. The Chunk Adapter connects to the output of the bottommost Transformer layer and used for identify if a token is the last token of a chunk. The QK Adapter is in parallel with Q and K matrix at each Transformer layer. It maps full attention scores to chunk attention scores, and trained by a distillation approach.

The QK Adapter fulfills feature compression and the generation of chunk attention scores. To train the QK Adapter, we propose an attention distillation approach designed to enhance the recall rate of key chunks. During training, LLM parameters are kept frozen, with the Kullback–Leibler (KL) divergence between chunk attention scores and full attention scores serving as a guidance signal for optimization. The Chunk Adapter determines whether a token corresponds to a chunk boundary by leveraging contextual semantic information. During the inference phase, we exploit the Intra-Chunk Attention Consistency (ICAC) pattern such that chunk selection is only updated when the current token is identified as a chunk boundary, which substantially enhances inference efficiency. Furthermore, ChunkLLM can achieve inference performance comparable to that of models optimized for 120K context lengths, despite being trained solely on 4K context lengths, thereby substantially reducing the training overhead associated with 120K context scaling. Experimental results validate that ChunkLLM yields a 4.48× speedup relative to the vanilla Transformer when processing 120K long texts.

Our contributions are summarized as follows:

- We introduce ChunkLLM that integrates two lightweight and pluggable components into existing LLMs: the QK Adapter and the Chunk Adapter. The newly developed ChunkLLM only necessitates fine-tuning these lightweight components on the basis of the original model architecture. This design enables ChunkLLM to attain performance comparable to vanilla Transformer while utilizing a smaller KV cache, alongside achieving effective control over computational scale.

- We propose an attention distillation-based training approach for the QK Adapter, which leverages KL divergence to drive chunk attention toward approximating full attention, effectively enhancing the recall rate of key chunks. Furthermore, we introduce a novel ICAC pattern, which yields notable improvements in inference efficiency for long-context scenarios.

- Experimental evaluations show that ChunkLLM not only attains comparable performance on short-text benchmarks but also maintains 98.64% of the performance on long-context benchmarks while preserving a 48.58% key-value cache (kvcache) retention rate, relative to the vanilla Transformer. Particularly, ChunkLLM attains a maximum speedup of 4.48× in comparison to the vanilla Transformer in the processing of 120K long texts.

## 2 METHOD

The framework of ChunkLLM is shown in Figure 1. ChunkLLM can be built on any existing transformer-based LLMs. Two extra lightweight and pluggable modules are designed to support chunk-related capability. One is Chunk Adapter, which is used to identify chunk boundaries. The

other is the Q Adapter and K Adapter, which is tailored for efficient feature compression and chunk selection. This section elaborates on the details of the two modules.

## 2.1 CHUNK ADAPTER

The Chunk Adapter is a one-layer forward neural network (FNN) classifier for chunk boundary prediction. Its input is the output of the first layer of the LLM, and the output is if or not the token is a chunk boundary, as depicted in the Figure 1.

For an input $X = \{x_1, x_2, ..., x_{n-1}, x_n\}$ with $n$ tokens and their corresponding labels $Y = \{y_1, y_2, ..., y_{n-1}, y_n\}$, $y_i \in \{0, 1\}$, where 1 indicates that the token $x_i$ is a chunk boundary, 0 indicates that it does not, $\mathbf{H}_i^{l_1}$ be the output of $x_i$ at first layer. The FNN based Chunk Adapter is given as in equation 1

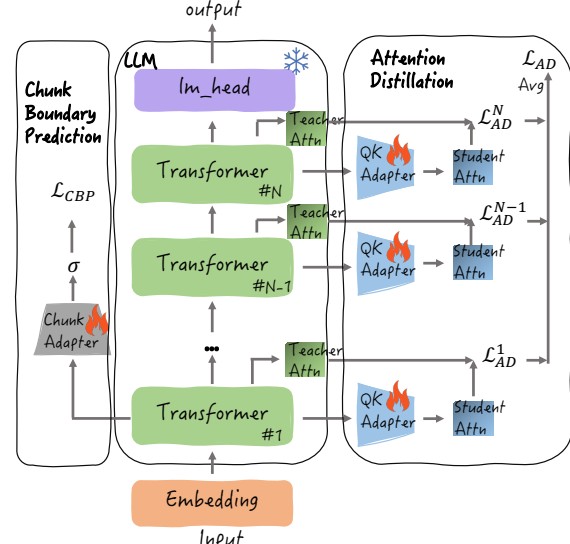

$$\hat{y}_i = \begin{cases} 1, & \text{Sigmoid}(\text{FFN}(\mathbf{H}_i^{l_1})) > \alpha, \\ 0, & otherwise \end{cases} \tag{1}$$

Figure 1: The framework of ChunkLLM.

For training the chunk adapter, we employ the binary cross-entropy loss (BCE) (equation 2) as the objective function. Detailed information on the training dataset will be given in the experiment part.

$$\mathcal{L}_{CBP} = -\frac{1}{n} \sum_{i=1}^{n} [y_i \cdot log(\hat{y}_i) + (1 - y_i) \cdot log(1 - \hat{y}_i)] \tag{2}$$

## 2.2 QK ADAPTER

At each layer of the LLM, we incorporate a Q-Adapter and a K-Adapter which used to compress the attention and select the chunks.

For each layer, let $\mathbf{Q}$ and $\mathbf{K}$ be the attention matrix respectively, $c$ be the chunk number of the input, $Index\_c = \{i_1, i_2, ...i_c\}$ is the index set of chunk boundary tokens. Let $\hat{\mathbf{K}}$ be the K matrix of these tokens. We then calculate chunk attention scores as follows:

$$\mathbf{A}^s = Softmax(\frac{Mul(\bar{\mathbf{Q}}, \bar{\mathbf{K}}^T)}{\sqrt{d_k}})$$

$$\bar{\mathbf{Q}} = FFN_Q(\mathbf{Q}), \bar{\mathbf{K}} = FFN_K(\hat{\mathbf{K}}) \tag{3}$$

where $FFN_Q$ and $FFN_K$ is proposed Q-Adapter and K-Adapter, respectively, $\bar{\mathbf{Q}} \in \mathbb{R}^{n \times d_k}, \bar{\mathbf{K}} \in \mathbb{R}^{c \times d_k}$, $d$ is the dimension of the model, and $d_k$ is the dimension of head. $d_k \ll d$.

**Attention Distillation** We propose an attention distillation strategy to train the Q-Adapter and K-Adapter. Where, we treat $\mathbf{A}^s$ as student attention, and a type of aggregation of original attention $\mathbf{A}^t$ which is given in follow as teacher attention. The objective is to align the student's chunk attention with that of the teacher, improving the recall performance for key chunks. As shown in Figure 1.

For the sake of descriptive simplicity, we use a single head as an illustrative example to show how to aggregate original attention. The calculation procedure is detailed as follows:

$$\mathbf{A}^t = Aggregate(\mathbf{A})$$

$$\mathbf{A} = Softmax(\frac{Mul(\mathbf{Q}, \mathbf{K}^T)}{\sqrt{d_k}}) \qquad \mathbf{A} \in \mathbb{R}^{n \times n} \tag{4}$$

where $\mathbf{Q} \in \mathbb{R}^{n \times d_k}$ and $\mathbf{K} \in \mathbb{R}^{n \times d_k}$ are the matrices of query and key for one attention layer. For brevity, the mask operation is omitted from the description.

$Aggregate$ denotes the operation of summing the token scores within a single chunk. Assuming that an input comprises $c$ chunks with $n$ tokens. Under this setting, $A_{ij}^t$ denotes the attention score of the current token $x_i$ relative to $j\text{-}th$ chunk. For multi-head attention, we compute the average along the head dimension, yielding matrix $\mathbf{A}^t$.

We employ the Kullback-Leibler (KL) divergence as the loss function for attention distillation to guide the student model $\mathbf{A}^s$ in approximating the teacher model's attention scores $\mathbf{A}^t$ :

$$\mathcal{L}_{AD}^N = KL(\mathbf{A}^t || \mathbf{A}^s) \tag{5}$$

We average the KL divergence losses across the N layers to obtain the final attention distillation loss:

$$\mathcal{L}_{AD} = \frac{1}{N} \sum_i^N \mathcal{L}_{AD}^i \tag{6}$$

During the training phase, the parameters of the backbone network are frozen, with only the Chunk Adapter and QK Adapter undergoing training, thereby achieving efficient training.

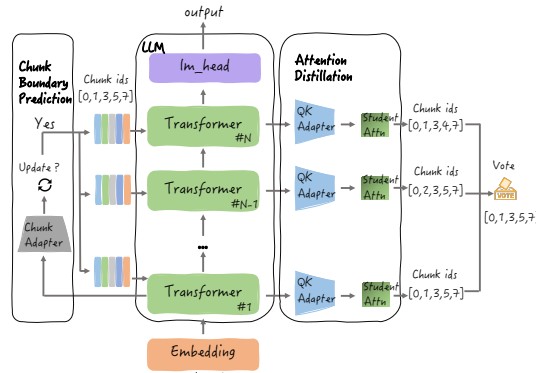

Figure 2: The inference process of ChunkLLM.

## 2.3 INFERENCE

The inference phase of ChunkLLM is depicted in Figure 2, encompassing two primary steps: top-k chunk selection and ICAC. In line with the ICAC paradigm, chunk updates are triggered exclusively when the current token functions as a chunk boundary.

**Top-k Chunk Selection** This stage is primarily dedicated to selecting top-k chunks for each layer. To elaborate, we use the first layer as an illustrative example and define $e$ as the end position of the input sequence. We then derive the attention scores $A_e^s = \mathbf{A}^s[e,:] \in \mathbb{R}^{1 \times c}$ that cor-

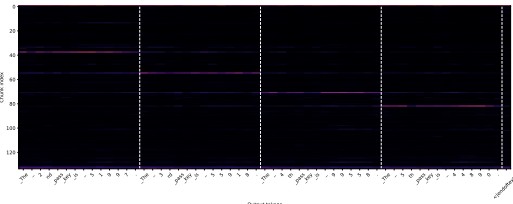

Figure 3: Attention visualization of chunk selection during the inference phase. The sample is derived from the passkey retrieval task.

respond to the $c$ chunks associated with the end token. $[,]$ denotes the slicing operation. We select the indices of the top-k chunks with the highest scores from $A_e^s$, where $k \ll c$, and retrieve the corresponding k chunks from $\mathbf{K}$ and $\mathbf{V}$, which facilitates the selection of the top-k key chunks. $\mathbf{V}$ are the value matrices for one attention layer. We propose a chunk voting mechanism that performs voting on the top-k chunks from each layer, thereby deriving the global top-k chunks. These retrieved chunks are subsequently stored in the KV-cache.

**ICAC** We find a phenomenon during the model inference, as illustrated in Figure 3. The chunks attended to by tokens within a generated chunk exhibit substantial consistency, whereas chunk updates predominantly occur at chunk boundaries. We name this phenomenon the "**I**ntra-**C**hunk **A**ttention **C**onsistency (ICAC)".

ICAC makes it possible to save computational cost in chunk selection. We incorporate the chunk boundary prediction task into the inference phase. Only when the currently decoded token is a chunk boundary, do we update the chunk selection and integrate the complete chunk from the prediction phase into **K** and **V**; otherwise, no update is executed.

# 3 EXPERIMENTS AND RESULTS

## 3.1 EXPERIMENTAL SETTINGS

### 3.1.1 MODEL AND BASELINES

Two representative open-source models, Qwen2.5-7B (Team, 2024) and Llama3.1-8B (Dubey et al., 2024), are chosen as the target models for evaluation. We select StreamingLLM (Xiao et al., 2024) and SepLLM (Chen et al., 2024) as the baselines to benchmark the proposed method. In detail, StreamingLLM retains both initial tokens and window tokens, whereas SepLLM, developed based on StreamingLLM, treats separator features as chunk features and incorporates a specialized separator cache management mechanism in the inference stage. Detailed settings of the experimental parameters are provided in Appendix 6.1.

### 3.1.2 TRAINING DATASETS

The FineWeb-Edu dataset (Lozhkov et al., 2024) is employed as the training corpus in this study. Developed by the HuggingFaceFW team, this dataset undergoes filtering via an educational quality classifier, which constructed based on annotations generated by Llama3-70B-Instruct (Meta, 2024).

For preprocessing the training data, the pySBD tool (Sadvilkar & Neumann, 2020), a rule-based sentence boundary detection module that works out-of-the-box, is utilized to annotate the end positions of chunks in sequences, serving as foundational input for training the chunk boundary prediction module.

### 3.1.3 BENCHMARKS

**Long Context Benchmarks** We select two long-context evaluation datasets, LongBench (Bai et al., 2024) and Needle In A Haystack (NIAH) (Kamradt, 2023), to assess the model's long-context ability The average text length for most tasks ranges from 5k to 15k tokens in LongBench. We select 10 of its subtasks for evaluation. Comprehensive information regarding the characterization of subtasks, evaluation methodologies, and additional relevant details is available in Appendix 6.2. For NIAH, the benchmark constructs prompts for LLMs by randomly inserting key information into long texts. The primary objective of this test is to verify whether large models can successfully extract such embedded key information from long context, thereby gauging the models' proficiency in long-context information extraction.

**Short Context Benchmarks** The selection of the evaluation datasets is primarily centered on the model's performance in three key dimensions: **General Knowledge**, which evaluates the model's breadth of knowledge coverage and the accuracy of its knowledge, MMLU (Hendrycks et al., 2021) (5-shot), SciQ (Welbl et al., 2017) (5-shot), OpenBookQA (Mihaylov et al., 2018) (25-shot); **Question Answering**, which evaluates the model's capabilities in question understanding and information matching, CommonsenseQA (Talmor et al., 2019) (5-shot), Social IQA (Sap et al., 2019) (15-shot), PIQA (Bisk et al., 2020) (25-shot); and **Reasoning** which evaluates the model's capabilities in logical abstraction and complex decision-making, HellaSwag (Zellers et al., 2019) (10-shot), Wino-Grande (Sakaguchi et al., 2020) (25-shot), ARC-c/ARC-e (Clark et al., 2018) (25-shot).

## 3.2 MAIN RESULTS

### 3.2.1 RESULTS ON LONGBENCH

We set the top-k ratio to 45% and the number of local chunks to 15 for ChunkLLM. The experimental results on the LongBench using Qwen2.5-7B and Llama3.1-8B are presented in Table 1. Here, "StrmLLM" represents StreamingLLM (Xiao et al., 2024). We take Qwen2.5-7B as an example for analysis, and the same conclusion holds for Llama3.1-8B. The following observations can be made:

| Methods | SDQA | | | MDQA | | | Summary | | Few-shot | | Avg | KV |
|---|---|---|---|---|---|---|---|---|---|---|---|---|
| | NQA | Qasper | MFQA | HQA | Musi | 2WQA | GR | QMS | SAM | TREC | | |
| Qwen2.5-7B | 33.76 | 51.40 | 57.17 | 56.76 | 29.40 | 38.09 | 31.57 | 24.40 | 46.73 | 72.00 | 44.13 | 100.00 |
| ChunkLLM | **31.09** | **50.52** | **56.95** | **55.61** | 29.41 | **38.27** | **31.25** | **23.16** | 46.53 | **72.50** | **43.53** | 48.58 |
| SepLLM | 26.43 | 50.18 | 50.29 | 47.25 | 22.83 | 36.34 | 28.31 | 21.14 | 46.71 | 71.50 | 40.10 | 53.17 |
| StrmLLM | 28.88 | 50.37 | 50.47 | 51.60 | 24.55 | 37.76 | 30.36 | 22.27 | **46.84** | 72.00 | 41.51 | 68.50 |
| Llama3.1-8B | 40.88 | 51.72 | 58.47 | 52.68 | 35.80 | 42.52 | 30.64 | 24.44 | 47.10 | 73.00 | 45.73 | 100.00 |
| ChunkLLM | **39.14** | 49.93 | 53.45 | **52.29** | **34.40** | **42.98** | **31.05** | **23.89** | **47.20** | 72.50 | **44.68** | 50.18 |
| SepLLM | 36.23 | **51.16** | 51.81 | 47.70 | 27.83 | 40.85 | 27.12 | 21.84 | 46.89 | 72.00 | 42.34 | 54.56 |
| StrmLLM | 35.94 | 50.96 | **54.13** | 50.54 | 30.95 | 41.83 | 27.50 | 23.06 | 46.23 | **73.00** | 43.42 | 69.25 |

Table 1: Experimental results on LongBench. Avg denotes average score. SDQA: single-document question answering, MDQA: multi-document question answering. The full names of the subtasks are shown in Appendix 6.2.

(1) In terms of overall average performance, ChunkLLM attains the optimal performance when compared to SepLLM and StreamingLLM, with respective improvements of 3.43 and 2.02 (43.53 v.s. 40.10 v.s. 41.51). In contrast to the short-text benchmark in Subsection 3.2.4, ChunkLLM demonstrates a remarkable improvement in long-context evaluation, which validates the advantage of ChunkLLM in retrieving key chunk information during long-context reasoning. (2) Notably, in the MDQA task, ChunkLLM yields a substantial improvement over SepLLM. We argue that the core challenge of MDQA lies in the dispersion of critical information across distinct positions within the context, which places high demands on the model's context comprehension capability. SepLLM leverages separators as chunk features, which is plagued by constrained representational capacity and the problem of chunk semantic incompleteness. By contrast, ChunkLLM enriches the representational capacity of chunks via attention distillation, which enhances the recall rates of critical chunks. This, in turn, effectively boosts the model's long-context understanding capability. (3) ChunkLLM attains 98.64% of the vanilla model's performance while employing the minimum KV cache. Notably, relative to SepLLM and StreamingLLM, ChunkLLM reduces the KV cache usage rate by 4.59% and 19.92% (48.58% v.s. 53.17% v.s. 68.50%), respectively, findings that further substantiate the superiority of ChunkLLM in long-context scenarios.

### 3.2.2 RESULTS ON NIAH

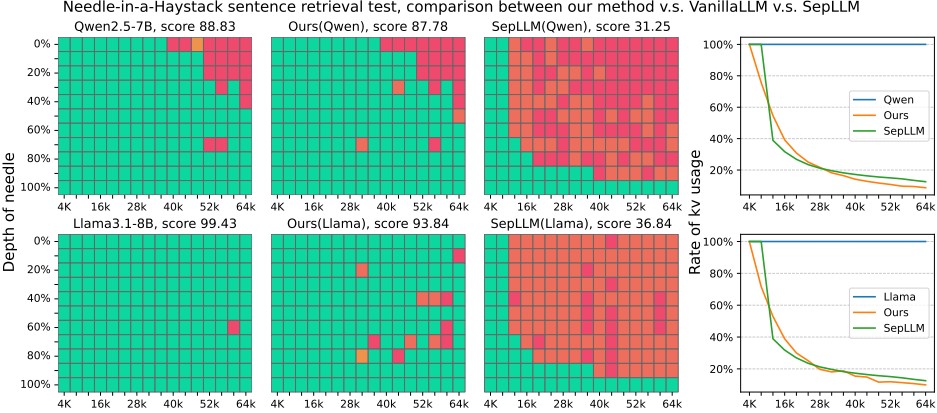

Figure 4: Needle-in-a-Haystack retrieval accuracy across context positions with 64k context length. The last column represents the KV-cache utilization rate.

We set the top-k to 256 and the number of local chunks to 16 for ChunkLLM. As depicted in Figure 4, ChunkLLM outperforms SepLLM across all scenarios in the 64K-context NIAH evaluation conducted on Qwen2.5-7B and Llama3.1-8B, achieving superior performance. Notably, in scenarios where the context length exceeds 12K, SepLLM exhibits near-total loss of retrieval capability (visualized in red), whereas ChunkLLM retains performance comparable to the vanilla model. This discrepancy is primarily attributed to ChunkLLM's attention distillation mechanism, which strengthens the feature representational capacity of chunks. Consequently, during chunk selection,

the model effectively identifies critical chunks with higher query relevance, leading to improved inference performance. Additionally, ChunkLLM exhibits a reduced KV-Cache utilization rate relative to SepLLM, which further corroborates the effectiveness of key chunk retrieval. We also conduct experiments with StreamingLLM, as shown in Appendix 6.3.

### 3.2.3 INFERENCE EFFICIENCY

Figure 5: Comparison of inference time per 10k tokens in the generation process on PG19 test set.

We conduct runtime evaluations of Vanilla and ChunkLLM for 120K-token generation tasks on the PG19(Rae et al., 2019) test set, with metrics recorded every 10K tokens. We set the top-k to 256 and the number of local chunks to 16 for ChunkLLM. As shown in Figure 5, as the number of generated tokens increases, Vanilla's inference time rises linearly, while ChunkLLM maintains persistent stability in time consumption. In the 110K–120K token generation phase, ChunkLLM outperforms Vanilla by speedups of 3.84× and 4.48×, which corroborates the efficacy of the proposed ICAC mechanism. During ChunkLLM's inference phase, chunk updates occur exclusively at chunk boundaries, minimizing the updates frequency and thereby boosting inference efficiency. We also conduct supplementary experiments using the FineWeb-Edu dataset, from which 1000 test corpora of 4k length are sampled. For the task of chunk boundary prediction, we evaluate its performance using three key metrics: precision, recall, and F1-score. The calculated results are 98.31, 95.54, and 96.91, respectively. Such promising performance indicators serve to verify the reliability and effectiveness of our chunk boundary prediction task.

We conduct experiments where these models generated 120K tokens, evaluating both total inference time and average perplexity (ppl) on the PG19 test set, and results are summarized in Table 2. Compared to the vanilla model, ChunkLLM yields a slight enhancement in ppl alongside a significant decrease in total inference time. The underlying reason is that while the vanilla model maintains semantic integrity, it incurs linearly increasing inference time as generated token count rises. Conversely, ChunkLLM reduces computational burden and speeds up inference by leveraging its chunk selection and ICAC mechanisms.

| Length | Methods | PPL | Total Time(s) |
|--------|---------|-----|---------------|
| 120K | Qwen2.5-7B | 14.41 | 10,684.31 |
| | ChunkLLM | 16.23 | 4,782.62 |
| | Llama3.1-8B | 11.93 | 14,906.19 |
| | ChunkLLM | 12.89 | 5,963.83 |

Table 2: The perplexity and running time comparison on the PG19 test set.

### 3.2.4 RESULTS ON SHORT TEXT

| Methods | General Knowledge | | | Question Answering | | | Reasoning | | | | Avg | KV |
|---------|------|------|------|------|------|------|------|------|------|------|------|------|
| | MMLU | SciQ | OQA | CQA | SIQA | PIQA | Heag | WG | ARC-c | ARC-e | | |
| Qwen2.5-7B | 74.25 | 97.00 | 52.80 | 84.52 | 58.44 | 81.72 | 80.24 | 77.27 | 63.82 | 87.21 | 75.73 | 100.00 |
| ChunkLLM | 72.51 | 96.60 | **52.40** | **84.68** | **58.34** | **81.77** | 80.08 | **76.87** | 63.65 | **87.21** | **75.41** | 45.47 |
| SepLLM | 73.07 | **96.70** | 52.20 | 84.19 | 58.25 | 81.41 | **80.10** | 76.48 | 62.94 | 86.11 | 75.15 | 50.20 |
| StrmLLM | **73.31** | 96.60 | 52.00 | 84.28 | 58.19 | 81.39 | 79.76 | 76.64 | 62.79 | 86.36 | 75.13 | 45.14 |
| Llama3.1-8B | 65.30 | 97.60 | 48.00 | 74.28 | 54.04 | 83.19 | 81.76 | 80.03 | 57.85 | 84.55 | 72.66 | 100.00 |
| ChunkLLM | **64.78** | 97.30 | **48.40** | **74.45** | **54.76** | 82.75 | **81.84** | **79.01** | 57.68 | **84.55** | **72.55** | 45.04 |
| SepLLM | 64.32 | **97.40** | 47.40 | 74.10 | 54.25 | **83.03** | 81.68 | 79.01 | **57.93** | 84.09 | 72.32 | 50.32 |
| StrmLLM | 61.19 | 97.20 | 48.00 | 73.79 | 53.94 | 81.56 | 80.14 | 78.22 | 56.91 | 83.59 | 72.45 | 45.30 |

Table 3: Experimental results on short context benchmarks.

The experimental results for short texts are presented in Table 3. The following conclusions can be drawn: (1) The overall average metrics of ChunkLLM on Qwen2.5-7B and Llama3.1-8B both outperform those of StreamingLLM and SepLLM, achieving 99.57% and 99.84% of the Vanilla model's performance, respectively. Notably, ChunkLLM attains optimal performance across 8 out of the 10 evaluation tasks, validating its efficacy in short-text task scenarios. (2) We perform statistical analyses on the average utilization rate of the KV cache. In comparison with SepLLM, ChunkLLM achieves superior performance while consuming a lower volume of KV cache (45.47% v.s. 50.20%). Specifically, on the Llama3.1-8B model, ChunkLLM not only exhibits the minimal KV cache usage but also outperforms both SepLLM and StreamingLLM in terms of performance metrics. This finding further validates the precision of ChunkLLM in chunk recall, achieving a balanced trade-off between performance and memory consumption.

## 3.3 Ablation Study

### 3.3.1 Effectiveness of Vote and ICAC

| Methods | SDQA | | | MDQA | | | Summary | | Few-shot | | Avg |
|---|---|---|---|---|---|---|---|---|---|---|---|
| | NQA | Qasper | MFQA | HQA | Musi | 2WQA | GR | QMS | SAM | TREC | |
| Qwen2.5-7B | 33.76 | 51.40 | 57.17 | 56.76 | 29.40 | 38.09 | 31.57 | 24.40 | 46.73 | 72.00 | 44.13 |
| ChunkLLM | 31.09 | 50.52 | **56.95** | **55.61** | 29.41 | 38.27 | 31.25 | 23.16 | 46.53 | **72.50** | 43.53 |
| w/o vote | 30.78 | 45.00 | 51.53 | 54.96 | 28.00 | 38.17 | 30.82 | 23.49 | 46.58 | 70.00 | 41.93 |
| w/o ICAC | **31.36** | **50.94** | 56.84 | 55.61 | 28.50 | **38.59** | **31.86** | **23.93** | **46.66** | 72.50 | **43.68** |

Table 4: Ablation Study on LongBench, w/o vote: remove vote mechanism, w/o ICAC: remove ICAC pattern.

We validate the proposed vote mechanism and ICAC pattern based on the Qwen2.5-7B using the LongBench, with experimental results shown in Table 4. Removal of the vote mechanism leads to a 1.6 drop in overall performance (43.53 v.s. 41.93), which confirms the mechanism's efficacy, as it integrates inter-layer differences in chunk selection and minimizes interference arising from such discrepancies. We also conduct a visual investigation into the recall performance of top-k chunks across differ-

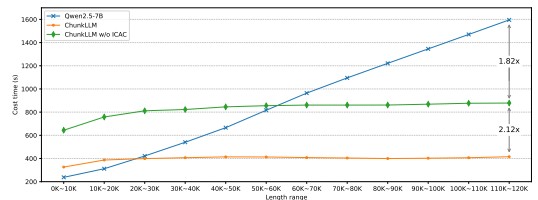

Figure 6: Inference efficiency of ICAC on PG19 test set.

ent layers, with comprehensive experimental results provided in Appendix 6.5. Conversely, removing ICAC results in a marginal 0.15 improvement in overall performance. This slight gain is attributed to the increased frequency of chunk selection updates during the inference phase. Frequent chunk selection, however, poses a limitation of low inference efficiency. As shown in Figure 6, after ICAC is removed, the inference latency is 2.12 times higher than that of ChunkLLM in the 110K–120K token generation stage. Conversely, incorporating ICAC enables the model to maintain near-lossless performance alongside improved inference efficiency, which provides additional validation of ICAC's success. Appendix 6.4 shows a case study of the ICAC.

### 3.3.2 Analysis of Fixed Chunks and Semantic Chunks

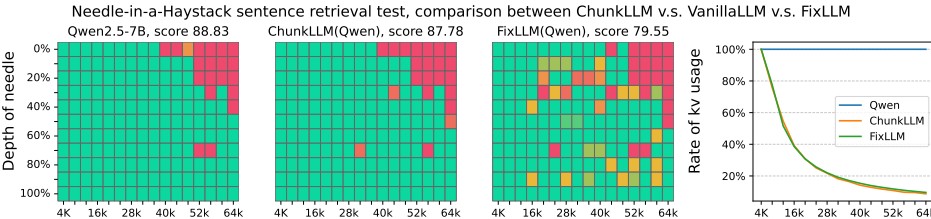

Figure 7: Visualization of fixed chunks and semantic chunks in NIAH test.

We conduct an experimental analysis of the fixed chunk method (FixLLM) on the NIAH task. To ensure consistent KV cache utilization and facilitate a fair comparison, FixLLM is configured with a top-k of 384 and a local chunks of 24, while ChunkLLM is set to a top-k of 256 and a local chunks of 16. The experimental results are illustrated in Figure 7. As observed, under conditions of approximately consistent KV cache utilization, FixLLM exhibits an 8.23 reduction (87.78 v.s. 79.55) in accuracy relative to ChunkLLM on the 64K NIAH task. This discrepancy stems from the semantic incompleteness of fixed chunks, which in turn compromises chunk selection during the inference phase. In contrast, ChunkLLM leverages contextual semantic information to identify chunk boundaries, preserving the semantic integrity of chunks.

## 4 RELATED WORK

**KV Cache Compression** Recent research has primarily focused on overcoming the limitations of Large Language Models (LLMs) in processing massive contextual inputs. SnapKV (Li et al., 2024) improves efficiency through KV cache compression, using attention scores to select and cluster important positional information; H2O (Zhang et al., 2023) implements a dynamic token retention policy that balances recent information and historically important information to optimize memory occupancy; StreamingLLM (Xiao et al., 2024) enables LLMs to handle sequences of infinite length without fine-tuning by retaining attention sinks and local tokens; PyramidInfer (Yang et al., 2024) and PyramidKV (Cai et al., 2024) optimize performance by adjusting the KV cache capacity across different layers However, most methods in this category cannot be applied to the training phase.

**Sparse Attention** The sparse attention mechanism constructs sparse attention matrices by confining attention to predefined patterns, such as local windows or fixed-stride block patterns. Beltagy et al. (2020a) combined dilated local window attention with task-specific global attention. MoBA (Lu et al., 2025) proposes an innovative mixed block attention mechanism. ESA (Wang et al., 2025) reduces computational costs by selecting tokens most critical to the current generation for attention calculation. NSA (Yuan et al., 2025) combines coarse-grained token compression and fine-grained token selection. SepLLM (Chen et al., 2024) finds that the segment information between separators can be effectively compressed into the separators themselves without causing significant information loss.

**Knowledge Distillation** Knowledge Distillation (Hinton et al., 2015), as a widely used model compression technique, aims to train a student model under the guidance of a teacher model (Rusu et al., 2016; Sanh et al., 2019; Gou et al., 2020). For text generation tasks, the standard KD method approximates the minimization of the forward Kullback-Leibler Divergence (forward KLD) between the generation distributions of the student and teacher models (Kim & Rush, 2016; Taori et al., 2023; Chiang et al., 2023; Peng et al., 2023a; Sanh et al., 2019).

## 5 CONCLUSION

We introduce ChunkLLM, a lightweight and pluggable framework, only necessitates fine-tuning lightweight components, QK Adapter and Chunk Adapter, on the basis of the original model architecture. And then we propose an attention distillation-based training approach for the QK Adapter, which leverages KL divergence to drive chunk attention toward approximating full attention, effectively enhancing the recall rate of key chunks. Furthermore, we introduce a novel "Intra-chunk Attention Consistency Pattern," which yields notable improvements in inference efficiency for long-context scenarios. Experimental results show that ChunkLLM attains a maximum speedup of 4.48× in comparison to the vanilla Transformer in the processing of 120K long texts.

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

## 6 APPENDIX

### 6.1 PARAMETER SETTING

The Qwen2.5-7B and Llama3.1-8B models are trained with identical configurations. A cosine annealing strategy is adopted, with a maximum learning rate of 3e-5 and a warm-up period of 500 steps. Additionally, we use Adam optimizer with parameters beta1 = 0.9 and beta2 = 0.99. For the Qwen2.5-7B and Llama3.1-8B, the dimensions of the QK Adapter and Chunk Adapter are set to 3584 and 4096, respectively. The total number of additional parameters is 14.7M and 21M, respectively. $\alpha$ is set to 0.5 for chunk boundary prediction task. The training dataset comprised approximately 6B tokens, and the training process are conducted on 32 H200 GPUs for around 11,300 steps. We set training epoch is 1, and the attention distillation stage consum 11 hours, while the chunk boundary training stage took 1.5 hours.

For a fair comparison on long-context benchmarks, we set the initial cache capacity to 4, the local cache capacity to 4000, separator cache capacity to 4000 and the maximum cache capacity to 8192 for SepLLM (Chen et al., 2024). For StreamingLLM (Xiao et al., 2024), we configure the initial cache capacity as 4 and the local cache capacity as 8188.

For short context benchmarks, we set the initial cache capacity to 4, the local cache capacity to 256, and the maximum cache capacity to infinity for SepLLM (Chen et al., 2024). For StreamingLLM (Xiao et al., 2024), we configure the initial cache capacity as 4 and the local cache capacity as 256.

### 6.2 SUBTASK DESCRIPTION ON LONGBENCH

Table 5 presents the task name, abbreviations, evaluation methodologies, average lengths, and descriptive details of each subtask on LongBench.

| Task | Subtask | Abbreviation | Evaluation | Avg Len | Description |
|------|---------|--------------|------------|---------|-------------|
| SDQA | NarrativeQA | NQA | Recall | 18,409 | Answer questions based on stories or scripts, including understanding of important elements such as characters, plots, themes, etc. |
| | Qasper | Qasper | Recall | 3,619 | Answer questions based on a NLP research paper, questions proposed and answered by NLP practitioners. |
| | MultiFieldQA-en | MFQA | Recall | 4,559 | Answer English questions based on a long article, which comes from a relatively diverse field. |
| MDQA | HotpotQA | HQA | Recall | 9,151 | Answer related questions based on multiple given documents. |
| | Musique | Musi | Recall | 11,214 | Answer related questions based on multiple given documents. |
| | 2WikiMultihopQA | 2WQA | Recall | 4,887 | Answer related questions based on multiple given documents. |
| Summary | GovReport | GR | Rouge-L | 8,734 | A summarization task that requires summarizing government work reports. |
| | QMSum | QMS | Rouge-L | 10,614 | A summarization task that requires summarizing meeting records based on user queries. |
| Few-shot | SAMSum | SAM | Rouge-L | 6,258 | A dialogue summarization task, providing several few-shot examples. |
| | TREC | TREC | Accuracy | 5,177 | A classification task that requires categorizing questions, includes 50 categories in total. |

Table 5: Task description on LongBench. SDQA: single-document question answering, MDQA: multi-document question answering.

### 6.3 COMPARE WITH STREAMINGLLM ON NIAH

As depicted in Figure 8, ChunkLLM outperforms StreamingLLM across all scenarios in the 64K-context NIAH evaluation conducted on Qwen2.5-7B and Llama3.1-8B with lower KV-cache usage, achieving superior performance. SteamingLLM leverages initial tokens and local tokens as its core token selection strategy. However, this design inherently limits its ability to effectively capture critical information situated in the middle segment of the input sequence, thereby resulting in a notable deficiency in long-context retrieval performance.

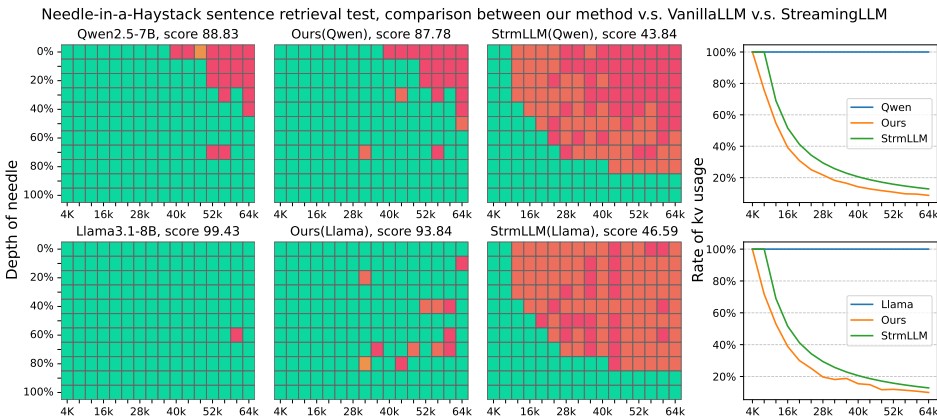

Figure 8: Compare with StreamingLLM on Needle-in-a-Haystack. The last column represents the KV-cache utilization rate.

## 6.4 CASE STUDY

To illustrate the reasoning process, we randomly sample one example from the LongBench summarization task, with its visualization results presented in Figure 9. As observed in the figure, during the generation phase, the chunks attended to by tokens within the same chunk demonstrate remarkably high consistency, corresponding to the highlighted bands in the visualization, whereas shifts in attention occur exclusively at chunk boundaries. This empirical observation validates the effectiveness of the proposed ICAC pattern.

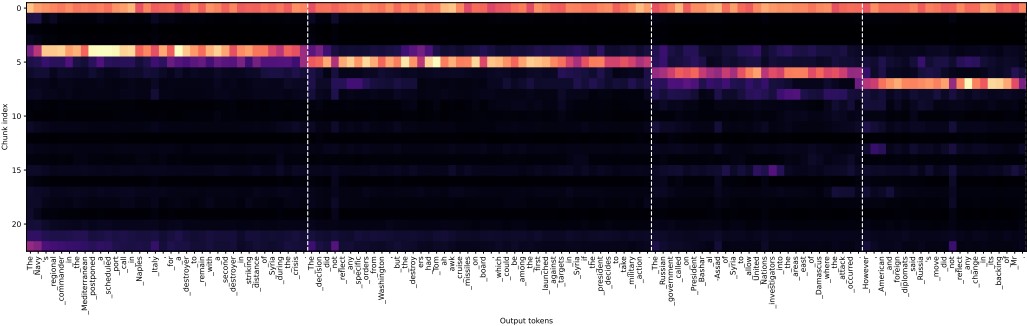

Figure 9: Case study of ICAC. The test case is from the LongBench summarization task.

## 6.5 TOP-K CHUNKS CROSS ALL LAYERS

Figure 10 presents the recall performance of top-k chunks across all layers based on Qwen2.5-7B. A consistent pattern emerges: the chunk recall rate in the lower layers (Layers 0–6) is relatively low, which we attribute to insufficient semantic representation. In contrast, the middle layers (Layers 7–20) demonstrate a notably higher chunk recall rate. Specifically, when top-k is set to 15, the recall rate of these middle layers exceeds 80%. This phenomenon, we contend, stems from the richer semantic representations inherent in the middle layers, coupled with the fact that our proposed attention distillation strategy effectively enhances the model's chunk selection capability. Conversely, the chunk recall rate in the highest layers (Layers 21–27) exhibits a downward trend; we attribute this to the functional role of the highest layers, which are primarily dedicated to facilitating the model's output generation. Notably, the chunk voting mechanism can effectively mitigate discrepancies between cross-layer chunks, and thereby enables the achievement of optimal performance.

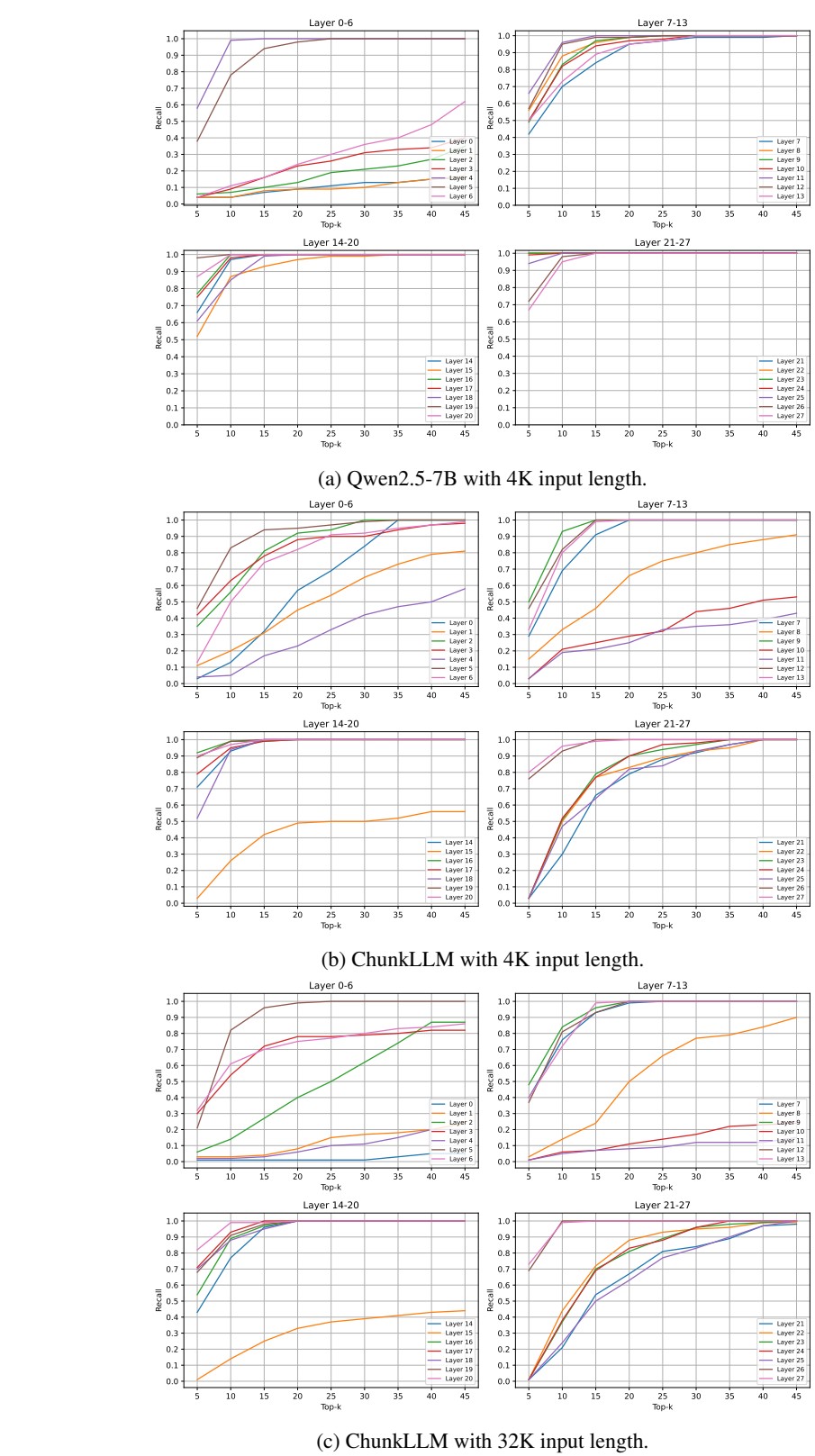

(a) Qwen2.5-7B with 4K input length.

(b) ChunkLLM with 4K input length.

(c) ChunkLLM with 32K input length.

Figure 10: Top-k chunks aross all layers.

