# OpenReview forum: "ChunkLLM: A Lightweight Pluggable Framework for Accelerating LLMs Inference"
_ICLR.cc/2026/Conference — ICLR 2026 Conference Withdrawn Submission_

### Official Review · Reviewer_kRmt · 2025-10-28

**Soundness:** 1
**Presentation:** 3
**Contribution:** 2
**Rating:** 4
**Confidence:** 3

**Summary:**

This paper introduces ChunkLLM, a training and inference pipeline that employs a sentence-segmentation approach. It isolates the attention calculation for different sentences and uses a retrieval mechanism designed to compute attention only for the most relevant sentences.

**Strengths:**

Overall, the paper is comprehensive and clearly written. The experimental validation is thorough and supported by key visualizations.

**Weaknesses:**

In my opinion, while this paper is sufficiently novel, many of its designs are overly engineered and lack rigorous theoretical or empirical justification. Although this provides acceleration, it simultaneously introduces significant drawbacks: it fails to generalize to all task types, and its application scenarios are potentially ad-hoc.

**Questions:**

1.  How can the attention statistics of an entire chunk be estimated using only the separate token of that chunk? Theoretically, the relationship between the two is very distant, and a simple FFN seems insufficient for this process. Even with training, it is intuitively difficult to approximate, likely leading to inaccurate chunk retrieval. Could the authors explain this or provide convincing empirical evidence?

2.  Clearly, different layers, and even different heads within each layer, serve distinct functions. Intuitively, forcing all layers to vote should be significantly worse than allowing each layer to retrieve its own chunks. Why then, in the ablation study, did performance decrease after removing the vote mechanism? Furthermore, what is the motivation for this vote mechanism, and what advantages does it have over layer-specific retrieval?

3.  Regarding the ICAC property, is it related to the vote mechanism? It seems the vote mechanism inherently smoothes the chunk retrieval, causing all tokens within the same sentence to retrieve identical chunks. Do the authors agree with this assessment? Moreover, is this ICAC property present in the vast majority of examples, or does it only manifest in specific tasks like summarization or passkey retrieval?

4.  The sentence separation mechanism introduces calibration data, which prevents generalization across all datasets. For data types such as code and math, can the training method proposed by the authors still be effectively applied for adaptation?

5.  Why was LongBench not fully evaluated? I recommend the authors complete the evaluation on more subsets, especially the synthetic tasks.

---

> ### Author Response · Authors · 2025-12-02
>
> We sincerely appreciate your in-depth questions, which allow us to further elaborate on the motivation and efficacy of our design. Our point-by-point responses are outlined below.
> 1. Inspired by operations in works like SepLLM and NSA (where chunk semantics are compressed into a single token), we attempt to estimate per-chunk attention scores based on such a compressed token. As evidenced in the Appendix, our trained ChunkLLM (Qwen) attains a needle-chunk recall rate in key-retrieval that is nearly on par with the teacher model. This validates the effectiveness of our training approach.
> 2. We agree with you that different model layers exhibit distinct functionalities. Consistent with mainstream observations, we also found that lower layers lack pronounced sparse attention patterns, making attention distillation at these layers ineffective. Indeed, this is a known challenge in related works, most of which fall back to full attention computation in these layers to circumvent the issue. However, this approach incurs significant computational overhead. To resolve this more efficiently, we experimentally explored and identified the voting mechanism as an effective mitigation strategy. Consequently, we introduced voting into ChunkLLM, which allows us to maintain effectiveness without relying on ineffective distillation in lower layers, thereby further improving ChunkLLM.
> 3. The ICAC phenomenon is primarily linked to the chunking strategy rather than the voting mechanism. Our observations indicate that ICAC manifests across various tasks including summarization, retrieval, and QA, and is most pronounced in summarization and retrieval tasks. Therefore, we selected and presented the attention heatmaps specifically for these two representative task types in the paper.
> 4. We have supplemented the experimental results of ChunkLLM on code and mathematical tasks.
> - Code Tasks: As shown in the LCC and RB-P columns of the table in Point 5, ChunkLLM continues to perform strongly, demonstrating good generalization to this domain.
> - Mathematical Tasks: Performance on mathematical tasks (results below) shows a noticeable decline. To achieve satisfactory results, it is necessary to retain over 60% of the KV cache.
> We attribute this primarily to the lack of mathematical task data in our training set, which likely limits the model's specialization in this area. While this indicates that ChunkLLM's generalization is somewhat constrained in math domains, it remains capable of handling a broad spectrum of general-purpose tasks effectively.
>
> | Model | Qwen2.5-7B | k-rate=0.3, n=5 | k-rate=0.5, n=5 | k-rate=0.5, n=10 | k-rate=0.45, n=15 |
> | :--- | :--- | :--- | :--- | :--- | :--- |
> | **GSM8K (5-shot)** | 80.14 | 51.86 | 61.50 | 78.06 | 78.97 |
> | **Use KV Rate (%)** | 100.00 | 38.27 | 55.52 | 63.16 | 70.48 |
>
> 5. The complete experimental results on the LongBench benchmark are presented below. (k-rate=0.45, n=15)
>
> | Models | NQA | Qasper | MFQA | HQA | Musi | 2WQA | GR | QMS | MN | SAMS | TQA | TREC | PC | PR | LCC | RB-P | Avg |
> | :--- | :--- | :--- | :--- | :--- | :--- | :--- | :--- | :--- | :--- | :--- | :--- | :--- | :--- | :--- | :--- | :--- | :--- |
> | **Qwen2.5-7B** | 33.76 | 51.40 | 57.17 | 56.76 | 29.50 | 38.09 | 31.57 | 27.20 | 24.40 | 46.73 | 92.50 | 72.00 | 2.00 | 66.50 | 62.86 | 53.53 | 46.62 |
> | **ChunkLLM** | 31.09 | 50.52 | 56.95 | 55.61 | 29.41 | 38.27 | 31.25 | 27.07 | 23.16 | 46.53 | 92.50 | 72.50 | 3.00 | 68.00 | 62.72 | 53.50 | 46.38 |

---

### Official Review · Reviewer_X2Tm · 2025-10-29

**Soundness:** 2
**Presentation:** 2
**Contribution:** 2
**Rating:** 2
**Confidence:** 5

**Summary:**

This paper proposes ChunkLLM, a lightweight pluggable framework, featuring QK Adapter and Chunk Adapter to accelerate LLM inference.

**Strengths:**

1. The idea is straightforward and makes intuitive sense.
2. The results of LLaMA-8B and Qwen-7B are good.

**Weaknesses:**

1. The main concern I have with the paper is the practicality of the proposed adapters. The added complexity of training adapters for each model makes it much less scalable than existing training-free methods, such as StreamingLLM [1] and H2O [2], despite some accuracy performance improvements. To the best of my knowledge, this work falls more into `incremental' work with over-engineering techniques, where I fail to see any valuable insights for the ICLR communities.

2. How would the proposed method perform against prior training-free methods in terms of latency?

3. How would the proposed method compare with newer baselines, such as ChunkKV [3] and Quest [4]?

4. How would the proposed method perform on larger models, such as 13B/30B?

5. The presentation needs some improvements. For instance, Figures 1 and 2 could be merged into one figure; Figure 3 is very hard to see.

6. Some system-related works on KV cache compression are missing [5-7].

[1] Efficient Streaming Language Models with Attention Sinks, ICLR 2024.

[2] H2O: Heavy-Hitter Oracle for Efficient Generative Inference of Large Language Models, NeurIPS 2023.

[3] ChunkKV: Semantic-Preserving KV Cache Compression for Efficient Long-Context LLM Inference, NeurIPS 2025.

[4] Quest: Query-Aware Sparsity for Efficient Long-Context LLM Inference, ICML 2024.

[5] InfiniGen: Efficient Generative Inference of Large Language Models with Dynamic KV Cache Management, OSDI 2024.

[6] Keyformer: KV Cache Reduction through Key Tokens Selection for Efficient Generative Inference, MLSys 2024.

[7] ALISA: Accelerating Large Language Model Inference via Sparsity-Aware KV Caching, ISCA 2024.

**Questions:**

Please see the weaknesses.

---

### Official Review · Reviewer_n39F · 2025-11-01

**Soundness:** 3
**Presentation:** 3
**Contribution:** 3
**Rating:** 4
**Confidence:** 4

**Summary:**

This paper introduces ChunkLLM, an efficient framework designed to enhance the computational efficiency of LLMs when processing long sequences. The core challenge addressed is the complexity of the self-attention mechanism in Transformers. ChunkLLM proposes a solution by integrating two lightweight, trainable modules into pre-existing LLMs: a Chunk Adapter and a QK Adapter. The Chunk Adapter, attached to the first layer, performs chunk boundary prediction using a simple feed-forward network, thereby segmenting the input into semantically coherent units. The QK Adapter, operating in parallel at each Transformer layer, compresses the full attention map into a chunk-level attention score via a knowledge distillation process that minimizes the KL divergence between the full and chunk-based attention distributions. The authors demonstrate that ChunkLLM generalizes effectively to long contexts, achieving inference speedup over the vanilla Transformer while retaining its performance on long-context benchmarks.

**Strengths:**

(1) The problem discussed in the paper, improving the efficiency of transformer-based LLMs, is important and challenging.

(2) The design of the proposed method is nice. The proposed method is pluggable, which allows for direct integration into existing LLMs without requiring costly retraining of the entire model backbone.

(3) The paper provides a thorough evaluation across multiple dimensions. Experiments on long-context benchmarks, short-context tasks, and inference efficiency demonstrate the method's superiority over baselines. The ablation studies effectively validate the contribution of individual components.

**Weaknesses:**

(1) The paper selects StreamingLLM and SepLLM as baselines. However, it omits comparison against other long-context methods for the inference time evaluation, for example, H2O or SnapKV (dynamic KV cache compression methods cited in the related work).

(2) The hyperparameters top-k and the number of "local chunks" are set empirically. There is no sensitivity analysis showing how performance and efficiency scale with these parameters. The optimal k is likely dependent on the total context length and task, and the paper provides no guidance on how to select it for a new application, representing a significant practical limitation for deployment.

(3) The paper focuses on computational speedup and KV cache size. However, it does not discuss potential memory overheads. The chunk voting mechanism and the storage of chunk-level attention scores introduce additional memory footprints. It would be great if the authors can introduce more about the full memory overheads.

**Questions:**

(1) How does the performance of ChunkLLM degrade as the semantic granularity required for chunking becomes more complex? For instance, would it struggle with documents with rich structures, such as tables?

(2) The chunk voting mechanism is shown to be beneficial, but what is the computational cost of this cross-layer consensus? Is there a risk of this process becoming a bottleneck itself when the number of layers and chunks is very high?

---

> ### Author Response · Authors · 2025-12-02
>
> We sincerely thank you for your constructive feedback. We have conducted additional analyses to address the raised points. Our key responses are as follows:
> 1. We have incorporated SnapKV into the comparative experiments.
> - Comparison Setup & Results: Following its original paper, we configured SnapKV with c=4096, w=32, k=7 and the maxpool strategy. Experimental results on the LongBench benchmark (see table below) demonstrate that our method still holds a performance advantage.
> - Analysis of Efficiency: SnapKV’s efficiency stems from performing KV filtering only once at the end of the Prefill phase, with no further filtering during decoding. Consequently, it is primarily suited for long-input, short-output tasks. For tasks requiring sustained generation, its efficiency becomes comparable to the vanilla chunking approach. In contrast, our method is designed to be efficient across a wider variety of task scenarios.
> - Note on H2O: We did not include H2O in our main comparison because it is not compatible with GQA and FlashAttention-2, which are integral to our experimental setup for fair and efficient benchmarking.
>
> | Models | NQA | Qasper | MFQA | HQA | Musi | 2WQA | GR | QMS | MN | SAMS | TQA | TREC | PC | PR | Avg |
> | :--- | :--- | :--- | :--- | :--- | :--- | :--- | :--- | :--- | :--- | :--- | :--- | :--- | :--- | :--- | :--- |
> | **Qwen2.5-7B** | 33.76 | 51.40 | 57.17 | 56.76 | 29.50 | 38.09 | 31.57 | 27.20 | 24.40 | 46.73 | 92.50 | 72.00 | 2.00 | 66.50 | 44.96 |
> | **ChunkLLM** | 31.09 | 50.52 | 56.95 | 55.61 | 29.41 | 38.27 | 31.25 | 27.07 | 23.16 | 46.53 | 92.50 | 72.50 | 3.00 | 68.00 | 44.70 |
> | **SnapKV** | 31.68 | 50.56 | 56.26 | 56.11 | 28.19 | 37.64 | 28.95 | 26.77 | 24.51 | 46.44 | 92.50 | 72.00 | 1.50 | 66.00 | 44.22 |
>
> 2. We performed a sensitivity analysis for hyperparameters on the LongBench benchmark. The overall results are summarized in the table below. A finer-grained analysis indicates that the model exhibits weak sensitivity to parameter N but stronger sensitivity to parameter K. Notably, on the SDQA subtask, optimal performance is achieved when roughly 50% to 60% of the KV cache is retained. In contrast, performance on the Summary and Few-shot subtasks remains highly stable with respect to both parameters.
>
> **Qwen2.5-7B (Local-N=15)**
> | Top-k-Rate | 0.1 | 0.2 | 0.3 | 0.4 | 0.5 | 0.6 | 0.7 | 0.8 | 0.9 |
> | :--- | :--- | :--- | :--- | :--- | :--- | :--- | :--- | :--- | :--- |
> | **Avg. Score** | 40.46 | 41.88 | 42.74 | 43.13 | 43.57 | 43.78 | 44.02 | 44.10 | 44.14 |
>
> 3. The additional memory overhead introduced in this part is minimal. During inference, compared to the vanilla model, our method retains only the per-chunk token outputs from each K-Adapter layer. The shape of this tensor is (C, 128), where C is the number of chunks. For Qwen2.5-7B processing a 128K-length sequence, C is typically around 10K. This results in an extra memory consumption of approximately 70 MB (calculated as 10,240 × 128 × 2 bytes * 28 layers)， which accounts for only about 1% of the total KV Cache. Furthermore, the chunk-level attention score tensor has a shape of (1, C) and is discarded immediately after the top-k selection is performed, rendering its memory footprint negligible.
> 4. We tested our model on coding tasks (results below). Even though our training data for attention distillation and chunk boundary prediction contained no code samples, the model still performs remarkably well. Our inspection reveals that the model's chunking on code data largely follows line breaks. This is a reasonable approach because lines in code often correspond to syntactic or functional boundaries, which aligns with the goal of chunk-based processing.
>
> **(Top-K=0.45, Local-N=15)**
> | Model | LCC | RepoBench-P |
> | :--- | :--- | :--- |
> | **Qwen2.5-7B** | 62.86 | 53.53 |
> | **ChunkLLM** | 62.72 | 53.50 |
>
> 5. The computational overhead of the voting mechanism is negligible. The operation involves a simple counting selection applied to the Top-K results from each layer. Its time complexity is O(N × K log K), where N is the number of model layers and K is the number of selected chunks per layer. To quantify this, even if we select half of the chunks (K) for a 128K-length task, K typically remains under 5,000. For Qwen2.5-7B (N=28), the total cost is negligible. Therefore, the voting mechanism does not pose any risk of becoming an efficiency bottleneck.

---

### Official Review · Reviewer_WiMv · 2025-11-01

**Soundness:** 2
**Presentation:** 2
**Contribution:** 2
**Rating:** 4
**Confidence:** 3

**Summary:**

The paper presents ChunkLLM, a lightweight, pluggable framework designed to accelerate LLM inference by reducing the quadratic cost of self-attention. The approach integrate two components to the existing Transformer achitectures: Chunk Adapter that predicts chunk boundaries based on contextual semantics and QK Adapters that used to distill attention. During training, the backbone model remains frozen, and only adapters are optimized via KL-divergence–based distillation. They perform experiments on Qwen2.5-7B and LLaMA3.1-8B show that ChunkLLM maintains comparable accuracy on short-text tasks and 98.6% on long-context benchmarks while achieving up to 4.48× inference speedup and about half KV-cache usage reduction.

**Strengths:**

1. The two modular adapters can be easily attached to existing LLMs without retraining the full model. And the functions of the two adapters are clear.

**Weaknesses:**

1. The papers does not have formal analysis of why attention distillation preserves semantic completeness or guarantees ICAC stability. Same for analysis of the chunk boundaries.
2. The paper is not compared with some current state-out-of-the art efficiency model such as FlashAttention 2, RingAttention, Mamba variants.
3. The selection for parameters like top-k chunk seem empirically tuned.
4. The figures look hand drawn which is a little bit weird.

**Questions:**

see weakness section

---

> ### Author Response · Authors · 2025-12-02
>
> We sincerely thank you for your engaged reading and thoughtful questions. These comments provide an excellent opportunity for us to clarify several key aspects of our work. In this response, we address the concerns point-by-point:
> 1. Semantic completeness and ICAC stability are governed by chunk boundary prediction training, not attention distillation. Inappropriate boundaries (e.g., splitting a coherent sentence) degrade these metrics. Our experiments show that PySBD is an effective sentence segmentation tool, and models trained on its outputs achieve robust chunking performance on most conventional texts.
> 2. Our method is fully compatible with FlashAttention-2, which was used for evaluating all models in the paper. Our study focused on comparisons within the Transformer architecture. Therefore, Mamba, representing a different architectural paradigm, was not a primary baseline for this work.
> 3. The hyperparameters were selected empirically. To validate this choice, we conducted a sensitivity analysis on the LongBench benchmark for parameters K and N. The results (provided below) confirm that our chosen values lie within a stable performance plateau, supporting their appropriateness.
>
> **Qwen2.5-7B (Local-N=15)**
> | Top-k-Rate | 0.1 | 0.2 | 0.3 | 0.4 | 0.5 | 0.6 | 0.7 | 0.8 | 0.9 |
> | :--- | :--- | :--- | :--- | :--- | :--- | :--- | :--- | :--- | :--- |
> | **Avg. Score** | 40.46 | 41.88 | 42.74 | 43.13 | 43.57 | 43.78 | 44.02 | 44.10 | 44.14 |
>
> **Llama3.1-8B (Local-N=15)**
> | Top-k-Rate | 0.1 | 0.2 | 0.3 | 0.4 | 0.5 | 0.6 | 0.7 | 0.8 | 0.9 |
> | :--- | :--- | :--- | :--- | :--- | :--- | :--- | :--- | :--- | :--- |
> | **Avg. Score** | 40.27 | 42.43 | 43.48 | 44.17 | 45.01 | 45.14 | 45.63 | 45.73 | 45.75 |
>
> 4. We appreciate the suggestion and will revise the figure styles accordingly to improve consistency and clarity.

---

### Note · Authors · 2026-01-05

I have read and agree with the venue's withdrawal policy on behalf of myself and my co-authors.